# High-Intensity Training for Knee Osteoarthritis: A Narrative Review

**DOI:** 10.3390/sports11040091

**Published:** 2023-04-20

**Authors:** Domiziano Tarantino, Tine Theysmans, Rosita Mottola, Jonas Verbrugghe

**Affiliations:** 1Department of Public Health, University of Naples Federico II, 80131 Naples, Italy; mottolarosi@gmail.com; 2REVAL—Rehabilitation Research Center, Faculty of Rehabilitation Sciences, Hasselt University, 3500 Hasselt, Belgium; tine.theysmans@student.uhasselt.be (T.T.); jonas.verbrugghe@uhasselt.be (J.V.)

**Keywords:** high intensity training, knee osteoarthritis, exercise therapy, rehabilitation, exercise rehabilitation

## Abstract

Knee osteoarthritis (OA) is the most common joint disease worldwide. Exercise therapy has been identified as a first-line treatment option in patients suffering from knee OA. High-intensity training (HIT) is an innovative exercise modality with potential in improving various disease-related outcomes. The purpose of this review is to explore the impact of HIT on knee OA symptoms and physical functioning. A comprehensive search of scientific electronic databases was conducted to identify articles on the effects of HIT on knee OA. Thirteen studies were included in this review. Ten compared the effects of HIT with those of low-intensity training, moderate-intensity continuous training, or a control group. Three evaluated the effects of HIT alone. Eight reported a decrease in knee OA symptoms (especially pain), and eight reported an increase in physical functioning. HIT was shown to improve knee OA symptoms and physical functioning, but also aerobic capacity, muscle strength, and quality of life with minimal or no adverse events. However, compared with other exercise modalities, no clear superiority of HIT was found. HIT is a promising exercise strategy in patients with knee OA; nonetheless, the actual quality of evidence remains very low, and more high-quality studies are needed to confirm these promising outcomes.

## 1. Introduction

Osteoarthritis (OA) is the most common joint disease worldwide, affecting approximately 15% of the worldwide population [1]. Moreover, the incidence of symptomatic OA is on the rise due to aging of the world population and the global obesity epidemic [2]. OA is a leading cause of disability and can affect people’s physical and mental well-being. Symptoms are joint pain and stiffness with relevant consequences on functional status, significantly restricting daily activities and often resulting in a reduction of quality of life (QoL) [3,4]. Knee OA is the most prevalent form of OA [5], and it is a leading cause of disability among older people, with recent data affirming that over 560 million people are living with knee OA worldwide [6].

Physical activity is a safe, cost-effective, and recommended first-line knee OA treatment option for the management of pain and mobility for affected patients [7,8,9,10,11,12,13,14]. Exercise has been shown to have effect sizes comparable to those for simple analgesics and nonsteroidal anti-inflammatory drugs [15]. Unfortunately, few people with knee OA achieve recommended physical activity levels (i.e., 150 min/week of moderate intensity) [16,17,18,19], mainly demonstrating sedentary or inadequate physical activity behaviours [16,17]. Furthermore, people with knee OA have the added barriers of pain and functional limitations that make the recommended quantity of exercise intolerable [18,19].

Clinical guidelines recommend strength and aerobic training for patients with knee OA based on clinical trial evidence of effectiveness [9,15,20,21,22]. Aerobic training can promote the metabolism of adipose tissue, prevent muscle atrophy, accelerate the recovery of damaged cartilage, enhance the body’s immunity, and reduce pain [23]. Strength training mediates pain relief [24], enhances psychological well-being [25], maintains cartilage integrity in animal models [26,27], and may increase the shock absorbing capability of lower extremity muscles during walking [28]. Although exercise has been recognized as a core treatment for knee OA [29], it is still unclear which program is more effective [30]. One promising modality of therapeutic exercise in those with knee OA could be represented by high-intensity training (HIT) [31].

High-intensity training is generally defined as a percentage of more than 60% to 80% of the one repetition maximum (1 RM) in case of resistance or strength training [32,33,34], or a target intensity between 80% and 100% of maximal heart rate (HRmax) in case of cardiopulmonary exercise [35].

HIT programs have been shown to enhance exercise capacity, muscle strength, and health-related parameters compared with training programs at lower intensity [36]. Similar to healthy people, HIT also substantially improved these variables in chronic musculoskeletal disorders, such as axial spondyloarthritis [37], chronic non-specific low back pain [32,38,39,40,41,42,43], chronic neck pain [44], and other chronic disorders such as multiple sclerosis [45], heart failure [46] and cardiometabolic disease [35], and chronic obstructive pulmonary disease [47], as well as decreasing the level of disability caused by these disorders.

HIT programs such as high-intensity interval training (HIIT) and high-intensity resistance training efficiently improve exercise capacity and a wide range of health-related outcomes [36,39,48,49]. For example, HIIT has been proposed as a time-efficient form of exercise that may overcome motivational barriers associated with traditional moderate intensity training [50], showing promising long-term adherence rates and similar physiological benefits as less intense long-duration exercise in a shorter period and with more pronounced effects on cardiorespiratory fitness [36,51,52,53,54].

Previous studies have provided conflicting evidence regarding the impact of HIT on knee OA symptoms. Some suggested that HIT may be detrimental for knee OA symptoms due to the greater contact forces exerted on the joint [55], and that it might aggravate symptoms such as pain and swelling [56], although others did not support these findings [57,58]. In contrast, one study even suggested that short-term, high-intensity strength training is in fact safe and well-tolerated by older adults with knee OA [59]. Preliminary evidence from a small-sample study has suggested that high-intensity resistance training may have beneficial effects on muscle strength compared with low-intensity resistance training in patients with knee OA [56].

Furthermore, one of the working mechanisms of HIIT was found to be the activation of the sirtuins [60]. These are anti-inflammatory and anti-aging genes that turn on when the body is under certain types of stress or perceived stress, such as high-intensity exercises. It has recently been found that downregulation of sirtuin 1 activation is linked with age-related health issues, including OA [61].

Given these premises, the aim of this review was to understand what the benefits of HIT in patients with knee OA are compared with other exercise modalities or no physical therapy regarding knee OA symptoms and physical functioning, and to what extent are these benefits superior to other exercise modalities. To address the lack of available literature on the effects of HIT on knee OA, the authors conducted a narrative review using a systematic approach.

## 2. Materials and Methods

A comprehensive search of scientific databases, including PubMed, Scopus, and Web of Science, was performed by two independent authors (D.T. and J.V.) to collect relevant articles on the topic.

The selection of the available articles was made as reported in the flowchart in Figure 1.

Two independent reviewers (T.T. and J.V.) extracted and evaluated the data, and then designed the tables.

The authors also evaluated the reference lists of the included articles but eventually found no extra articles to be included. A list of the excluded studies is reported in the Appendix A.

The included articles reported on the effects of HIT without a comparator group or HIT compared with other forms of training (such as low-intensity or moderate-continuous), or HIT compared with control, on knee OA, and reported primary or secondary outcomes related to knee OA symptoms, muscle strength, and physical or functional status. Reported improvements were defined as statistically significant for *p*-values < 0.05.

Furthermore, to ensure the training protocols were considered as HIT by the authors of the present review, articles explicitly labelling their interventions as HIT and/or reporting exercise protocols for knee OA performed at an intensity regarded as HIT by the definitions reported above [32,33,34,35] were included, and specific keywords including “high-intensity training”, “high-intensity training AND osteoarthritis”, and “high-intensity training AND osteoarthritis AND knee” were used during the search.

To facilitate understanding of the results, we categorized the studies into the following sections: HIT (alone), HIT vs. low-intensity training (LIT), HIT vs. moderate-intensity continuous training (MICT), HIT vs. control (CT), and HIT vs. LIT vs. CT. We recommend that readers refer to the individual articles for additional details regarding the training protocols.

Finally, the selected studies were evaluated using the CERT tool (Table 1), which is designed specifically for the reporting of exercise programs across all evaluative study designs for exercise research [62].

## 3. Results

Our research results lead to the selection of 13 articles. A summary of the characteristics of the included studies is provided in Table 2.

### 3.1. HIT (without a Comparator Group)

Three studies reported on the outcomes of HIT alone for knee OA.

A pilot study by Golightly et al. [63] assessed the feasibility and changes in outcomes of a HIIT program in patients with symptomatic knee OA. Twenty-nine participants were enrolled to a 12-week (2×/week) supervised HIIT program. The authors stated that their HIIT program improved the Western Ontario and McMaster Universities Osteoarthritis Index (WOMAC) scores (32.4 ± 14.0, 28.8 ± 19.2, and 20.0 ± 13.7 at baseline, 6 weeks, and 12 weeks, respectively), 20-m fast-paced walk test measured in seconds (12.5 ± 5.4, 11.5 ± 5.0, 10.0 ± 2.8 at baseline, 6 weeks, and 12 weeks, respectively), 30-s chair-stand test measured in repetitions (12.6 ± 5.4, 14.5 ± 6.1,16.4 ± 6.4 at baseline, 6 weeks, and 12 weeks, respectively), stair-climb test measured in seconds (13.8 ± 7.3, 11.5 ± 6.4, 10.6 ± 6.2 at baseline, 6 weeks, and 12 weeks, respectively), timed up and go test measured in seconds (7.9 ± 4.8, 7.3 ± 4.4, 6.0 ± 1.4 at baseline, 6 weeks, and 12 weeks, respectively), balance measured as single leg stance in seconds (15.1 ± 11.0, 19.2 ± 11.6, 21.0 ± 11.9 at baseline, 6 weeks, and 12 weeks, respectively), isometric knee extensor strength measured in Newton meter (Nm) (average right knee 71.1 ± 33.4, 72.5 ± 36.5, 77.8 ± 37.9 at baseline, 6 weeks, and 12 weeks, respectively; average left knee 68.3 ± 35.0, 72.2 ± 37.3, 72.1 ± 33.5 at baseline, 6 weeks, and 12 weeks, respectively), and cardiorespiratory fitness, with most changes occurring as early as six weeks.

Another pilot and feasibility study by Smith-Ryan et al. [31] examined the feasibility of a 6-week HIIT program in patients with symptomatic OA. Sixteen participants were enrolled to a 6-week (12 exercise sessions + two testing sessions at baseline + two post-test sessions) HIIT program (2×/week). The authors stated that their HIIT program improved cardiorespiratory fitness and OA symptoms measured with WOMAC (pre-testing total score 36.15 ± 8.60; post-testing at six weeks 25.46 ± 16.09) in concert with metabolic alterations indicative of improved skeletal muscle energetics.

The last study on HIT alone for knee OA was performed by King et al. [57] to evaluate the effects of a HIT knee extensor and flexor resistance training program on strength, pain, and adherence in patients with advanced knee OA and varus malalignment. Fourteen patients with medial compartment knee OA and malalignment were enrolled to a 12-week (3×/week) HIT program. The authors concluded that their supervised HIT training produced substantial increases in knee extensor and flexor strength (strength increased from 28% to 46%, relative to baseline values) in middle-aged patients with advanced knee OA and varus malalignment, without concomitant increases in pain, adverse events, or decreases in adherence. Adherence was generally good throughout the studies (the drop-out rate was 27.59%, 18.75%, and not defined in the last article, respectively), and no adverse events related to the training programs were reported.

### 3.2. HIT vs. MICT

Only one study by Keogh et al. [64] evaluated the feasibility and potential effects of HIIT cycling as an alternative exercise option to MICT cycling for patients with knee OA. Twenty-seven participants were enrolled to an 8-week (4×/week) HIIT or MICT program. Significant improvements in health-related QoL measured with WOMAC (pre- and post-test scores for the HIIT group 36.1 ± 15.0 and 34.8 ± 15.5, respectively, pre- and post-test scores for the MICT group 21.2 ± 14.6 and 22.9 ± 14.4, respectively) were found for both groups, with the HIIT group also reporting significant increases in functional performance as assessed using the Timed Up and Go measured in seconds (pre- and post-test 8.9 ± 2.0 and 7.8 ± 1.1, respectively) and Sit to Stand measured in seconds (pre- and post-test 11.1 ± 2.2 and 13.1 ± 2.7, respectively) functional tests. The enrolment rate and adherence rate for both the HIIT and MICT groups was comparable to other cycling studies involving similar populations [72,73,74], with a withdrawal rate of 37%. Nonetheless, the number of adverse events (HIIT: two of nine participants, 26 adverse events; MICT: one of eight participants, two adverse events) appeared to be very high, although 24 of these adverse events were reported by one HIIT patient with a Bakers cyst on their knee, who still completed 25 of the requested 32 exercise sessions.

### 3.3. HIT vs. LIT

Three studies evaluated the outcomes of HIT vs. LIT for knee OA.

One recent study by de Zwart et al. [65] assessed whether resistance HIT led to increased muscle strength compared with resistance LIT in patients with knee OA. One-hundred and seventy-seven participants were enrolled to a 12-week (3×/week) HIIT or LIT program. In both HIT and LIT groups, muscle strength measured in Nm per kilogram (Nm/kg) (0.98 ± 0.40 and 1.11 ± 0.40 at baseline and 12 weeks, respectively, for the HIT group; 1.02 ± 0.41 and 1.15 ± 0.42 at baseline and 12 weeks, respectively, for the LIT group), knee pain measured with the Numeric Rating Scale (4.8 ± 2.3 and 2.9 ± 2.0 at baseline and 12 weeks, respectively, for the HIT group; 5.1 ± 2.4 and 2.7 ± 2.4 at baseline and 12 weeks, respectively, for the LIT group), and WOMAC-Physical Functioning (21.3 ± 13.0 and 16.8 ± 12.2 at baseline and 12 weeks, respectively, for the HIT group; 20.4 ± 13.3 and 16.4 ± 12.6 at baseline and 12 weeks, respectively, for the LIT group) were significantly increased after 12 weeks of training and at follow-up compared to the start of the training. Interestingly, researchers found higher estimated 1 RM strength values favoring HIT. Furthermore, significant differences were found in the Hospital Anxiety and Depression scale between the HIT and LIT groups in favor of the HIT group. The authors concluded that HIT did not result in greater improvements in isokinetic muscle strength, pain, and physical functioning compared with LIT in patients with knee OA; however, it was well-tolerated, so they suggested that either intensity of resistance training could be utilized in exercise programs for patients with knee OA.

In a study by Foroughi et al. [66], the authors hypothesized that HIT progressive resistance training would improve lower limb dynamic alignment and function (lower knee adduction moment, increased muscle strength, and fewer knee OA symptoms). Fifty-four women with knee OA were enrolled into a 6-month (3×/week) HIT or LIT exercise program. Dynamic alignment and knee adduction moment did not change over time or between groups. Muscle strength improved in both groups over time, but significantly more in the HIT group (overall relative change in strength was 52.5% in the HIT and 33.0% in the LIT group). By contrast, gait speed measured in meters per second (m/s) (1.1 ± 0.17 and 1.2 ± 0.17 at baseline and six months, respectively, for the HIT group; 1.1 ± 0.19 and 1.2 ± 0.17 at baseline and six months, respectively, for the LIT group) and pain measured with WOMAC (5.7 ± 3.3 and 3.83 ± 2.7 at baseline and six months, respectively, for the HIT group; 6.7 ± 3.5 and 5.5 ± 3.6 at baseline and six months, respectively, for the LIT group) improved over time in both groups. Improvements in shank adduction angle were related to improvements in self-reported disability, but not to changes in muscle strength, gait velocity, or pain. Although muscle strength improved significantly more in the HIT group, the hypothesized reduction in knee adduction moment and shank and knee adduction angles were not evident after either exercise modality.

Mangione et al. [67] evaluated the effects of HIT and LIT stationary cycling on functional status, gait, overall and acute pain, and aerobic capacity. Thirty-nine adults with knee OA were randomized to either the HIT or LIT exercise group for 10 weeks (3×/week) of stationary cycling. The authors concluded that participants with knee OA in both groups improved in timed chair rise measured in seconds (23.31 ± 9.10 and 19.11 ± 6.62 at baseline and after 10 weeks, respectively), in the distance walked in six minutes measured in meters (489.59 ± 109.16 and 533.78 ± 104.99 at baseline and after 10 weeks, respectively), in the range of walking speeds measured in m/s (1.04 ± 21 and 1.05 ± 20 at baseline and after 10 weeks, respectively, for slow walking; 1.59 ± 33 and 1.67 ± 33 at baseline and after 10 weeks, respectively, for fast walking), in the amount of overall pain relief (in 70% of training sessions, subjects reported that pain decreased immediately after cycling), and in aerobic capacity measured at the treadmill GXT test in minutes (10.98 ± 3.95 and 13.17 ± 4.21 at baseline and after 10 weeks, respectively), with no differences between HIT and LIT groups. Therefore, the authors stated that the intensity of exercise did not have a differential effect on these outcomes. The authors finally highlighted that the improvements in function and aerobic capacity demonstrated in their study suggested that training was more than a “practice effect,” because testing was carried out using a variety of walking-based measures, and training was performed by stationary cycling. Adherence was generally very good throughout the studies (the drop-out rate was of 6%, 9%, and 7.8%, respectively). Interestingly, two minor adverse events during testing occurred in the LIT group in the study by Foroughi et al. [66], and two minor adverse events occurred during the testing and training in the study by Mangione et al. [67], but it was not specified in which group.

### 3.4. HIT vs. CT

Four studies evaluated the effects of HIT vs. CT for knee OA.

Bressel et al. [14] quantified the efficacy of a HIIT aquatic treadmill exercise program on measures of pain, balance, function, and mobility in patients with knee OA. Eighteen participants were enrolled into a 6-week HIIT protocol (2–3×/week). The participants involved in the study first completed a 4-week non-exercise control period followed by a 6-week aquatic treadmill exercise program that incorporated a balance and HIIT training component. The authors observed that patients with OA displayed reduced joint pain measured with the Knee Injury and Osteoarthritis Outcome Score’ (KOOS) subscale, with scores being 30–49% greater at six weeks than at pre-test evaluation, reduced usual pain values (at six weeks being 213% lower than the pre-test), improved balance measured with the sensory organization test equilibrium and strategy scores (values after the 6-week intervention being 10 and 2.5% greater than baseline, respectively), improved function measured with the sit-to-stand test (rising index scores improved from 0.49 ± 0.19% at baseline to 0.33 ± 0.11% after six weeks), and better mobility measured with walking speed in seconds (8.6 ± 1.4 at baseline to 7.8 ± 1.1 after six weeks; 10% lower). The same improvements were not observed after a non-exercise control period. The authors suggested that aquatic treadmill exercise that incorporates high-intensity intervals was well-tolerated by patients with OA and seems to be effective at managing symptoms of OA.

Thorstensson et al. [68] tested the effects of a short-term HIT on self-reported pain, function, and QoL. Sixty-one middle-aged participants were randomized to HIT or CT groups. Thirty participants were enrolled to a 6-week (2×/week) HIT program. The authors stated that a 6-week high-intensity exercise program had no effect on pain or function in middle-aged patients with moderate to severe radiographic knee OA. However, in the HIT group, an improvement was seen at six weeks in the KOOS subscale QoL after six weeks compared with the CT group (40 ± 15 and 46 ± 21, respectively), and the difference between groups was still persistent at six months.

A randomized controlled trial (RCT) by Waller et al. [69] investigated the effects of a 4-month (3×/week) HIT aquatic resistance training program on body composition and walking speed in post-menopausal women with mild knee OA, immediately after the intervention and after a 12-month follow-up. Additionally, the influence of leisure time physical activity was also investigated. Patients were randomly allocated into one of the two arms of the study (HIT or CT). The authors found that the HIT aquatic resistance training program was effective at decreasing fat mass four months after the intervention, as well as improving walking speed calculated as m/s (1.74 ± 0.15, 1.83 ± 0.16, and 1.82 ± 0.14 for HIT at baseline, 4 months, and 12 months, respectively; 1.73 ± 0.17, 1.76 ± 0.17, and 1.77 ± 0.13 for CT, at baseline, 4 months, and 12 months, respectively) in post-menopausal women with mild knee OA. Furthermore, daily leisure time physical activity (recorded as any type of activity and self-perceived intensity of each activity, i.e., low, moderate, or high) over the 16-month period had a significant effect on fat mass loss but no effect on walking speed.

Calatayud et al. [70] evaluated the effectiveness of a 8-week (3/week) HIT preoperative resistance training program in patients waiting for total knee arthroplasty (TKA). Fifty patients were randomly allocated to the HIT group or CT. The main finding of this study was that high-intensity pre-operative training improved strength: isometric knee flexion measured in kg (9.2 and 9.1 at baseline, 9.4 and 4.4 three months after surgery for HIT and CT, respectively), isometric knee extension measured in kg (23.5 and 23.5 at baseline, 22.8 and 14.3 three months after surgery for HIT and CT, respectively), hip abduction measured in kg (7.3 and 7.2 at baseline, 7.8 and 5.0 three months after surgery for HIT and CT, respectively), active knee range of motion (knee flexion was 104.0° and 104.2° at baseline, 101.2° and 96.4° at three months after surgery for HIT and CT, respectively, while knee extension was 14.4° and 14.0° at baseline, 8.2° and 13.9° at three months after surgery for HIT and CT, respectively), and functional measures such as the timed up and go test measured in seconds (8.6 and 8.5 at baseline, 7.0 and 8.7 at three months after surgery for HIT and CT, respectively), as well as reduced pain measured with WOMAC (54.0 and 53.2 at baseline, 25.0 and 30.7 at three months after surgery for HIT and CT, respectively), and length of hospitalization in the early post-operative periods compared with CT. Adherence was generally very good throughout the studies (the drop-out rate was of 0%, 8%, and 2% after 16 weeks and 13% after 12 months, and 12%, respectively). No adverse events were reported throughout the studies.

### 3.5. HIT vs. LIT vs. CT

Two studies evaluated the effects of HIT vs. LIT vs. CT for knee OA.

A recent study by Messier et al. [71] was performed to determine whether strength HIT reduced knee pain and knee joint compressive forces more than strength LIT and more than CT in patients with knee OA. A total of 377 participants were randomized into strength HIT or LIT or CT. The exercise protocols for both HIT and LIT were performed for 18 months (3×/week). Among participants with knee OA, strength HIT did not significantly reduce WOMAC knee pain or knee joint compressive forces at 18 months compared with strength LIT or with an attention CT group. Some improvements were reported in the mean knee flexor strength that was statistically significantly greater in both exercise groups than in the CT group at 18 months measured in Nm (35.0 ± 20.1, 38.2 ± 20.3, and 38.1 ± 22.5 at baseline; 51.5, 52.6, and 43.8 at the 18-month follow-up for HIT, LIT, and CT, respectively), and in the proportion of participants using pain medication that declined across the 18-month intervention period (45%, 34%, and 55% for HIT, LIT, and CT, respectively), with no statistical difference among the groups at the 18-month follow-up.

Another study by Jan et al. [56] compared the effects of resistance strength HIT and strength LIT in elderly subjects with knee OA. One-hundred and two participants were randomized into HIT, LIT, and CT groups, and trained for eight weeks (3/week). The authors reported that both HIT and low-intensity resistance strength trainings reduced pain measured with the WOMAC pain subscale (8.5 ± 3.8 and 4.8 ± 3.5 in the HIT group at baseline and after eight weeks, 7.8 ± 3.3 and 4.8 ± 2.7 in the LIT group at baseline and after eight weeks, 8.3 ± 4.6 and 7.1 ± 3.4 in the CT group at baseline and after eight weeks) and improved function measured with the WOMAC physical function subscale (26.4 ± 9.0 and 14.7 ± 8.5 in the HIT group at baseline and after eight weeks, 26.1 ± 8.1 and 14.8 ± 9.2 in the LIT group at baseline and after eight weeks, 25.4 ± 11.3 and 22.5 ± 10.9 in the CT group at baseline and after eight weeks) in patients with knee OA. Although HIT strength training demonstrated effect sizes that consistently were slightly greater than those achieved with low-intensity resistance strength training, the differences in improvement between the HIT and low-intensity groups were not significant.

Adherence was generally good throughout the studies (the drop-out rate was 25% and 4%, respectively). There were 87 non-serious adverse events in the study by Messier et al. [71]: 53 in the HIT, 30 in the LIT, and 4 in the CT groups. Of those, 29 were related to the program: 20 in the HIT, 9 in the LIT, none in the CT groups. No adverse effects related to the program were reported in the study by Jan et al. [56].

## 4. Discussion

In our review, we examined the effects of HIT on knee OA as a standalone exercise modality or in comparison to training modalities with other exercise intensities such as MICT and LIT, or to CT. To the best of our knowledge, this is the first review that assessed the potential role of HIIT in mitigating symptoms associated with knee OA.

HIT showed good outcomes in improving pain, physical functioning, muscle strength, cardiorespiratory fitness, and QoL, when evaluated alone or compared with CT [14,31,57,63,68,69,70]. However, HIT was shown to have similar effects in improving pain, function, and strength when compared with other exercise intensities, especially LIT. The included studies utilized various exercise modalities ranging from resistance training to aquatic treadmill training, with all exercise intensities showing superiority over no exercise.

Exercise therapy is known to provide significant improvements to patients suffering from chronic musculoskeletal pain conditions (including OA). This improvement is thought to occur through different possible underlying mechanisms, including the reconceptualization of pain-related fears, a hypo-analgesic effect, and changes in the immune system, ultimately leading to better pain control, functional ability, and overall well-being [23,75,76].

According to Beckwée et al. [77], several explanatory models can be described for exercise-induced improvement of knee OA-related symptoms, which can be categorized into five main components: neuromuscular, peri-articular, intra-articular, psychosocial components, and general fitness and health. The authors suggest that the clinical benefits of exercise therapy observed in patients with knee OA are likely due to a combination of these underlying mechanisms, and that future exercise studies taking all possible pathways into consideration should help in providing more targeted exercise recommendations for patients suffering from knee OA.

Furthermore, Runhaar et al. [78] stated that an increase of upper leg strength, a decrease of extension impairments, and improvement in proprioception were identified as possible mediators in the positive association between physical exercise and OA symptoms.

Although several studies investigated optimal training parameters for resistance training in patients with knee OA, no uniform training dose can be proposed due to the great heterogeneity in training protocols limiting a direct comparison. In the included studies, resistance training treatment duration ranged from six weeks to 18 months and was performed using different kinds of equipment. For instance, de Zwart et al. [65], Thorstensson et al. [68], Waller et al. [69], and Messier et al. [71] used resistance bands in their protocols, resulting in improvements in knee OA symptoms, muscle strength, and physical functioning. Foroughi et al. [66] and Jan et al. [56] used machines such as leg press and leg extension/flexion, finding improvements in knee OA symptoms and muscle strength.

Regarding cardiorespiratory exercise training, the included studies mainly used cycling as a modality, and treatment duration ranged from 6 to 12 weeks. Golightly et al. [63] and Smith-Ryan et al. [31] found benefits in knee symptoms and cardiorespiratory fitness with cycling, as well as Keogh et al. [64], who found improvements in health-related QoL and physical function, and Mangione et al. [67] who found positive effects in physical function, overall pain, and aerobic capacity. In a recent review, Zeng et al. [23] stated that low-intensity aerobic exercise is better for patients with severe knee OA, while high-intensity aerobic exercise is more suitable for patients with mild knee OA. In contrast, for mild knee OA patients with chronic diseases, HIT was found to be better, so it might be suggested that physicians should choose the most appropriate treatment based on each individual patient’s health status.

### 4.1. Strengths and Limitations

A strength of this narrative review is the use of the CERT tool [62], which provides a systematic and clear display of all relevant exercise components, which supports the reproducibility of exercise modalities, and thus patient outcomes. CERT can be used in all types of exercise interventions, and has a good inter-rater agreement in musculoskeletal exercise interventions [79]. The main limitation of this review is the small number of relevant studies found in the scientific literature about the effects of HIT for knee OA. Furthermore, in the included studies, there was a great heterogeneity of exercise programs in terms of modality and duration, making it difficult to compare the effects of the exercise programs. However, both short-term and long-term programs showed that HIT is a feasible and effective strategy for lowering pain and improving function, with minimal to no adverse events. Finally, the lack of a risk of bias assessment represents another limitation of the present study.

### 4.2. Clinical Implications

Given the great variety in applicability of the HIT protocol, the authors recommend adapting the exercise modality to the patient’s preferences and available equipment of the physical therapist to maximize patient’s motivation and exercise adherence. Moreover, physical therapists, together with the patient, should choose between cardiorespiratory or resistance training, given the evidence that the application of one of the exercise modalities is superior to a mixed program in patients suffering from knee OA [80].

### 4.3. Future Recommendations

Given the promising results of the articles included in this review, more high-quality research should be performed for further understanding of the beneficial effects of HIT in patients with knee OA, since there is still too little knowledge about its effectiveness. The evaluation of the effects of a multimodal HIT program (i.e., resistance training in combination with cardiorespiratory training) [81], patient profiling in order to prescribe a tailored HIT program [82], and the evaluation of the effects of HIT on other important outcome measures in knee OA such as depression, inflammation, sleep quality, etc. [83], should be considered as topics of interest for future research.

## 5. Conclusions

In conclusion, it is possible to state that HIT showed promising results as a therapeutic exercise for patients with knee OA. However, the actual quality of evidence remains very low, and further high-evidence studies (such as RCTs) are needed to confirm the promising outcomes reported in this review.

## Figures and Tables

**Figure 1 sports-11-00091-f001:**
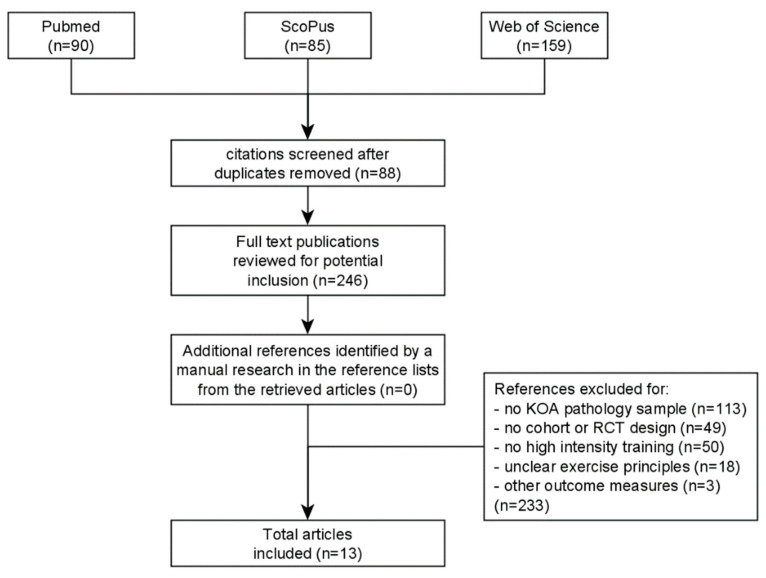
Flowchart illustrating the process of study selection.

**Table 1 sports-11-00091-t001:** Consensus on Exercise Reporting Template (CERT).

Study and Year	What (Equipment)	Who (Experienced Therapist?)	How						Where	When, How Much	Tailoring (of Exercise Intensity, Based on…)	How Well (Delivered and Performed as Expected?)
			Individual/Group (S/US)	Adherence/Adverse Events (Reported by...)	Motivation Strategies	Progression	Exercise Program Replicable/Home Program Available?	Non-Exercise Components				
Golightly, 2021 [63]	Cycle ergometer or treadmill; chest strap heart rate monitor	Yes	Individual (S)	Physiotherapist: number of sessions attended or completed/ participant + investigator	N.A.	N.A.	Yes, except exercise progression/N.A.		In center	20 min (without warm-up/ cooling-down); 10 reps of 1-min high-intensity bouts HI: 90% VO2peak Rest: 1 min complete rest between reps; 24 h between training sessions	Exercise modality; exercise intensity	Yes
Smith-Ryan, 2020 [31]	Electronically braked cycle ergometer	Yes	Individual (S)	Physiotherapist: number of sessions completed or total available/ participant + investigator	N.A.	N.A.	Yes, except exercise progression/ N.A.		In center	20 min (without warm-up/ cooling-down); 10 reps of 1-min high-intensity bouts HI: 90% peak power output Rest: 1 min between reps; 24 h between training sessions	Peak power output	Yes
King, 2008 [57]	Cycle ergometer; dynamometer (and accompanying software)	Yes	N.A. (S)	Researcher/ participant	N.A.	New targets of 60% peak torques based on test protocols every 3 weeks	Yes/N.A.		In center	45 min (including warm-up) Part 1 -> 3 sets; 10 reps; concentric isokinetic knee extension and flexion at 60, 90, and 120°/sec angular velocity; 60% of baseline strength Part 2 -> 3 sets; 15 reps; concentric isokinetic knee extension and flexion at 180°/sec angular velocity; maximum effort Rest: minimal 24 h between sessions	Peak torques	Minimal modification (due to increases in pain)
Keogh, 2018 [64]	Stationary bicycle	N.A.	Individual (US)	Participant (training diary)/ participant	N.A.	N.A.	Yes/yes		Home based	25 min HI: 7 min warm-up (progressively increasing intensity); 5 sets of high-intensity intervals at 110 rpm for 45 sec with 90 sec relative rest between sets at 70 rpm (low-intensity); 6–7 min cool-down (light-moderate intensity) Moderate-intensity: 3 min warm-up (light intensity); 20 min at 60–80 rpm (moderate intensity); 2 min cooldown (light intensity)	Personal experience	Yes
De Zwart, 2022 [65]	Fitness devices; weighted vests; ankle weights; resistance band	Yes	2×/week in group (S)	Physiotherapist/ participant + investigator	N.A.	↑ 5% 1-RM/ week, based on 1-RM measurement from week 0, week 6, and week 10	Yes/yes (paper hand-out)		2×/week in center	60 min; 3 sets; 10 reps HI: 70–80% 1-RM, weighted vests during weight-bearing exercises LI: 40–50% 1-RM, no additional weighted vests Rest: 90 sec between sets; 48 h between training sessions	% 1-RM	Yes
			1x/week individual (US)	Log book/ participant + investigator					1x/week home based			
Foroughi, 2011 [66]	Fitness devices	Yes	N.A. (S)	Researcher/ participant	N.A.	↑3% 1-RM	Yes/N.A.		In center	3 sets; 8 reps (6–9 sec/rep); 80% 1 RM (equal to 15–18 on the Borg Rating of Perceived Exertion) Rest: 10–15 sec between reps; 1–2 min between sets C: minimal resistance; no progression	% 1-RM	Main protocol deviation: changing to an isometric form of training if the dynamic mode was causing pain, reducing the intensity of the intervention group and/ or limiting the range of motion
Mangione, 1999 [67]	Cycle ergometer; heart rate monitor	N.A.	N.A. (S)	Researcher/ N.A.	N.A.	N.A.	Yes, except exercise progression/N.A.		In center	HI: 60 min (including warm-up and cool-down); 70% HR reserve LI: 60 min (including warm-up and cool-down); 40% HR reserve Target HR zone is reached by an increase in speed of pedaling instead of an increase in resistance	Maximum heart rate	Yes
Bressel, 2014 [14]	Pool; water jet; aquatic treadmill	N.A.	N.A. (S)	Researcher/participant + investigator	Verbal motivation	Progression visualized in table in study article	Yes/N.A.		In center	18–30 min; 3–6 reps HI: 14–19 RPE; 30 s–2 min 30 s Rest: 10 RPE; 1 min–1 min 30 s C: maintenance of typical activities of daily living; no new treatment	RPE	Yes
Thorstensson, 2005 [68]	Heart rate monitor (not obligatory); Thera-band	Yes	Group (S)	Researcher/N.A.	N.A.	Gradual increase in intensity by increased lever arm or range of motion	Yes/yes		In center + home based	Weight-bearing exercises; 1 h; minimal 60% maximum HR	Maximum heart rate	Yes
Waller, 2017 [69]	Pool; heart rate monitors; Thera-bands; Hydro-boots	Yes	Group (S)	Researcher/ investigator	N.A.	Progression from barefoot to small resistance fins and large resistance boots	Yes/N.A.		In center	1 h (i.e., 15 min warm-up, 30 min intensive aquatic resistance training program, 10–15 min cool-down); as hard and fast as possible C: maintenance of usual leisure time activities with the possibility to participate in 2 sessions of 1 h, including light stretching, relaxation, social interaction	RPE, heart rate, blood lactates	Yes
Calatayud, 2017 [70]	Fitness devices; Bosu^®^ Balance Trainer	Yes	N.A. (S)	N.A./ N.A.	N.A.	Adding a maximum of 2 or 3 kg	Yes/N.A.		In center	1 h (i.e., 15 min warm-up, 5 sets of 10 reps of each exercise, 5 min cool-down); 10-RM C: treatment including exercise which may explain the absence of between-group difference Rest: at least 48 h between sessions; 1 min between sets	1-RM	Yes
Messier, 2021 [71]	Nautilus resistance training machine; Thera-bands	Yes	Group (S)	Participant/ participant	Frequent contact; positive feedback; incentives; establishing personal commitment; promoting a sense of community	↑ 5% 1-RM/ 2 weeks, based on 1-RM measurement (measured each 9 weeks)	Yes/Yes		In center	60 min; 3 sets HI: 4–8 reps; 75–90% 1-RM LI: 15 reps; 30–40% 1-RM C: 60 min educational workshops and seated stretching 2x/month for the first 6 months and the other months 1x/month Rest: 60–90 sec between sets	% 1-RM	Yes
Jan, 2008 [56]	Leg press machine; bicycle (warm-up); cold pack (after exercise)	Yes	Individual (S)	Researcher/ N.A.	N.A.	↑5% 1-RM/ 2 weeks	Yes/N.A.		In center	HI: 30 min; 60% 1-RM; 3 sets; 8 reps LI: 50 min; 10% 1-RM; 10 sets; 15 reps C: no intervention Rest: 1 min between sets; 5 min between left and right knee sessions	%1-RM	Yes

Abbreviations. S: supervised; US: unsupervised; HI: High-Intensity; LI: Low-Intensity; reps: repetitions; 1-RM: One Repetition Maximum; HR: heart rate; rpm: revolutions per minute; I: intervention group; C: control group; 1-RM: One Repetition Maximum; N.A.: Not Available; RPE: Rate of Perceived Exertion.

**Table 2 sports-11-00091-t002:** Characteristics of included studies.

Study and Year	Exercise Modality	NOS; Age (Mean, Years); Sex (Female, n)	Groups	Exercise Duration & Sessions	Outcomes	Results	Drop-Out Rate
Golightly, 2021 [63]	CIT: cycling or walking	29; 63 ± 7; 19	I: High-intensity interval training (n = 29)	12 weeks; 24 sessions; 2×/week	Adverse events; feasibility; physical function; knee OA symptoms; balance; muscle strength; cardiorespiratory fitness; body composition	70% adherence No adverse events related to the program ↑ Physical function * ↓ Knee OA symptoms* ↑ Balance * ↑ Muscle strength * ↑ Cardiorespiratory fitness* Body composition: no significant changes	27.59%
Smith-Ryan, 2020 [31]	CIT: cycling	16; 59.9 ± 8.3; 12	I: High-intensity interval training (n = 16)	6 weeks; 12 exercise sessions + 2 testing sessions at baseline + 2 post-test sessions; 2×/week exercise sessions	Primary: feasibility Secondary: cardiorespiratory fitness (i.e., VO2peak); knee OA symptoms; circulating biomarkers of metabolism	Moderate feasibility: no adverse events related to the program, >96% adherence ↑ Cardiorespiratory fitness * ↓ Knee OA symptoms * ↓ Amino acids (i.e., methionine*, phenylalanine *, tyrosine*, serine) ↑ Aspartate/ asparagine ↓ Acylcarnitine	18.75%
King, 2008 [57]	RT: lower extremity	14; 48.35 ± 6.51; 2	I: High-intensity training (n = 14)	12 weeks; 36 sessions; 3×/week	Primary: knee extensor and flexor strength; pain; adherence Secondary: dynamic knee joint loading; patient-reported outcomes; self-efficacy after training	No adverse events ↑ Knee extensor and flexor strength * No increases in pain during or after training High adherence Dynamic knee joint loading and patient-reported outcomes: no significant changes ↑ Self-efficacy after training (i.e., function subscale) *	N.A.
Keogh, 2018 [64]	CIT: cycling	27; 62.4 ± 8.3; 13	I1: High-intensity interval training (n = 9) I2: Moderate-intensity continuous training (n = 8)	8 weeks; 32 sessions; 4×/week	Primary: feasibility (i.e., enrolment rate, withdrawal rate, exercise adherence, number of adverse events) Secondary: efficacy (i.e., health-related quality of life, physical function, body composition)	54% enrolment rate 37% rate of withdrawal Very high exercise adherence 28 adverse events (24 related to one HIIT participant) ↑ Health-related quality of life * ↑ Physical function * Body composition: no significant changes	37%
De Zwart, 2022 [65]	RT: lower extremity	177; 67.7 ± 5.8; 107	I1: High-intensity training (n = 89) I2: Low-intensity training (n = 88)	12 weeks; 36 sessions; 3×/week	Primary: isokinetic muscle strength; estimated 1-RM Secondary: knee pain; physical functioning; knee OA symptoms	No adverse events related to the program ↑ Isokinetic muscle strength ↑ estimated 1-RM ** ↓ Knee pain ↑ Physical functioning ↓ Knee OA symptoms	6%
Foroughi, 2011 [66]	RT: lower extremity	54; 64 ± 7; 54	I1: High-intensity training (n = 26) I2: Low-intensity training (n = 28)	6 months; 78 sessions; 3×/week	Primary: dynamic shank angles; knee adduction angles; knee adduction moment of the most symptomatic knee Secondary: muscle strength; gait speed; osteoarthritis symptoms	Two minor adverse events in the control group Dynamic shank angles: knee adduction angles, knee adduction moment: no change over time ↑ Muscle strength ** ↑ Gait speed * ↓Osteoarthritis symptoms *	9%
Mangione, 1999 [67]	Cardio-respiratory continuous training: cycling	39; 71 ± 6.9; 26	I1: High-intensity training (n = 19) I2: Low-intensity training (n = 20)	10 weeks; 30 sessions; 3×/week	Functional status; gait; overall and acute pain; aerobic capacity	↑ Functional status * ↑ Gait * ↓ Overall pain * No increase in acute pain ↑ Aerobic capacity *	7.8%
Bressel, 2014 [14]	CIT + balance training: aquatic treadmill	18; 64.5 ± 10.2; 16	I: High-intensity interval training (n = 18) C: Control group (n = 18)	High-intensity: 6 weeks; 2–3×/week Control: 4 weeks	Pain; balance; physical function; mobility	No adverse events related to the program ↓ Pain ** ↑ Balance ** ↑ Physical function ** ↑ Mobility **	0%
Thorstensson, 2005 [68]	RT: lower extremity	61; 56 ± 6; 31	I: High-intensity training (n = 30) C: Control (n = 31)	6 weeks; 12 sessions; 2×/week	Primary: self-reported pain; function; knee-related quality of life Secondary: health status; functional performance	Self-reported pain and function: no significant differences between groups ↑ Quality of life ** (at 6 weeks and 6 months) ↑ Health status ** (at 6 weeks) ↑ Functional performance*	8%
Waller, 2017 [69]	RT: aquatic	87; 63.8 ± 2.4; 87	I: High-intensity training (n = 43) C: Control (n = 44)	16 weeks; 48 sessions; 3×/week	Primary: body composition; walking speed Secondary: leisure time physical activity	High adherence ↓ Fat mass ** (after 4 months intervention) Lean mass: no significant changes ↑ Walking speed ** (after 4 months intervention and 12 months follow-up) Leisure time physical activity: significant effect on fat mass loss	2% (after 16 weeks) 13% (after 12-months follow-up)
Calatayud, 2017 [70]	RT: lower extremity	50; I: 66.8 ± 4.8, C: 66.7 ± 3.1; 37	I: High-intensity training (n = 25) C: Control (n = 25)	8 weeks; 24 sessions; 3×/week	WOMAC; SF-36; VAS; isometric knee flexion; isometric knee extension; isometric hip abduction; active knee range of motion; functional tasks	↓WOMAC ** ↑ SF-36 ** ↓VAS ** ↑ Isometric knee flexion ** ↑ Isometric knee extension ** ↑ Isometric hip abduction ** ↑ Active knee range of motion (i.e., flexion and extension) ** ↑ Functional tasks **	12%
Messier, 2021 [71]	RT: lower & upper extremity, core	377; 65; 151	I1: High-intensity training (n = 127) I2: Low-intensity training (n = 126) C: Control (n = 124)	18 months; 3×/week	Primary: knee pain; knee joint compressive force Secondary: physical function; mobility; disease progression; thigh skeletal muscle volume; thigh fat volume; IL-6 serum levels; knee extensor strength; hip abductor strength	29 nonserious adverse events related to the program No statistically significant difference between high-intensity and control group or high-intensity and low-intensity group	25%
Jan, 2008 [56]	RT	102; 63.3 ± 6.6 (high-intensity), 61.8 ± 7.1 (low-intensity), 62.8 ± 7.1 (control); 79	I1: High-intensity training (n = 34) I2: Low-intensity training (n = 34) C: Control (n = 34)	8 weeks; 24 sessions; 3×/week	Pain; function; walking time; muscle torque	↓ Pain * ↑Function * ↓Walking time * ↑Muscle torque * No statistically significant differences between the high-intensity and low-intensity groups	4%

Note. NOS: Number of subjects; RT: Resistance Training; CIT: Cardiorespiratory Interval Training; 1-RM: One Repetition Maximum; OA: osteoarthritis; IL-6: interleukin 6; VO_2_peak: peak oxygen consumption; N.A.: Not Available; WOMAC: Western Ontario and McMaster Universities Osteoarthritis Index; SF-36: Physical Functioning Scale of the Short Form-36; VAS: Visual Analogue Scale. * *p* < 0.05. ** significant difference between groups.

## Data Availability

Not applicable.

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
