# Peer review of "High-Intensity Training for Knee Osteoarthritis: A Narrative Review"

_sports, 2023, doi:10.3390/sports11040091_

Round 1

Reviewer 1 Report

First, I would like to recognize the authors for the preparation of this review article.

The title is precise and accurate.

The abstract presents the general findings of the reviewed studies on the impact of HIIT on knee osteoarthritis. A clear explanation was provided as to how many studies were analysed and the general findings of those were presented.

The introduction is clear and follows a logical sequence while all the relevant scientific support is provided. Lines 76-79: No need to use a different font here.

Table 1 should be modified. I was not able to follow what was included there.

A flow chart diagram of the study process could have been included.

Table 2 is long but nicely presented.

The results of the reviewed studies were presented in detail, discussed and interpreted! I find the comparison of the studies and discussion very helpful as the authors provide a detailed presentation of the reviewed studies.

Author Response

Dear Reviewer,

please find attached a point-to-point response to your comments.

Thank you.

Reviewer 2 Report

It is an interesting topic. Here are some comments.

One of the working mechanisms of HIIT is the activation of the sirtuins1. These are anti-inflammatory and anti-aging genes that turn on when the body is under certain types of stress or perceived stress, such as high-intensity exercises. It has recently been found that downregulation of sirtuin 1 activation is linked with age-related health issues, including OA2. I think the introduction or discussion section could be improved by mentioning this.

Why was there no meta-analysis or a priori published protocol? These limitations should be mentioned in the discussion section.

Please add a flow diagram illustrating the process of study selection.

Please add a table of studies excluded by full-text evaluation with reasons for exclusion and references or simply add the information in the results section. It can be put in the supplementary material.

Please state the methods for the literature search, risk of bias assessments, and data-extraction in under separate headings. Please remember to state the procedures, such as how many independent reviewers that completed the tasks and how. Please state relevant limitations in the discussion section.

In the method section, the authors stated that “… to ensure the training protocols were considered as HIIT by the authors of the present review, only articles explicitly labelling their interventions as HIIT were included …”. There should be a definition of HIIT in the review, and the authors should have used it in the search for relevant articles.

The readability of the manuscript is generally poor, and it makes it difficult to make a comprehensive peer-review. Here are a few examples:

In the abstract, the authors stated that “Nine compared the effects of HIIT with those of low-intensity training, moderate-intensity continuous training, or a control group.”. Please refer to the control group as “inactive control group” because there were many different types of control groups in the review. Please make this change in the entire manuscript.

In the abstract, it was stated that “Eight reported a decrease in knee OA symptoms, and eight reported an increase in physical functioning.”. It is unclear what the symptoms are and what the comparisons were.

The last sentence in the abstract needs to be revised or removed: “anyway, the actual quality of evidence remains very low, and more high-quality studies are needed to confirm these promising outcomes.”. What do the authors mean by quality of evidence? Please be specific.

The following text in the method section belongs in the background section: “To address the lack of available literature on the effects of HIIT on knee OA, the authors conducted a narrative review using a systematic approach.”.

The following text in the beginning of the results section belongs in the method or background section: “To facilitate understanding of the results, we categorized the studies into the following sections: HIIT (alone), HIIT vs low-intensity training (LIT), HIIT vs moderate-intensity continuous training (MICT), HIIT vs control (CT), and HIIT vs LIT vs CT. We recommend that readers refer to the individual 109 articles for additional details regarding the training protocols.”.

Please use the term “benefit” less often.

The text alignment in the tables needs fixing.

1. Gurd BJ, Perry CG, Heigenhauser GJ, et al. High-intensity interval training increases SIRT1 activity in human skeletal muscle. Appl Physiol Nutr Metab 2010;35(3):350-7. doi: 10.1139/h10-030 [published Online First: 2010/06/18]

2. Deng Z, Li Y, Liu H, et al. The role of sirtuin 1 and its activator, resveratrol in osteoarthritis. Biosci Rep 2019;39(5):BSR20190189. doi: 10.1042/BSR20190189

Author Response

Dear Reviewer,

please find a point-to-point response to your comments (reported in bold) in the attached file.

Thank you

Round 2

Reviewer 2 Report

The manuscript has been greatly improved. I suggest a minor revision before publication.

I still think a section with a description of the data-extraction procedures should be added. It is a systematic review with a narrative analysis of results, in my opinion. I can accept if only one reviewer extracted the data, but it needs to be clarified. Please mention the lack of two independent reviewers in the limitations section if this was the case.

When searching for eligible studies to be included in reviews, the reference lists of included studies should be read. Please mention the lack of this procedure in the limitations section.

The lack of a risk of bias assessment should also be mentioned in the limitations section.

Please write in past tense throughout the manuscript.

Author Response

Dear Reviewer,

please find attached a point-to-point response to your comments.

Best regards
